Applying a reservoir functional-zone paradigm to littoral bluegills: differences in length and catch frequency?

Ruhl Nathan 1 2 Ruhl@Rowan.edu
DeAngelis Holly 2
Crosby Abigale M. 2
Roosenburg Willem M. 2
1 Department of Biology, Rowan University , Glassboro, NJ , USA
2 Department of Biological Sciences, Ohio University , Athens, OH , USA
Kaushik Sadasivam
Electronic publication date: 2014 Aug 19
Publication date: 2014
Volume: 2
Electronic Location ID: e528
Received 2014 Feb 28; Accepted 2014 Jul 29
Copyright: © 2014 Ruhl et al.
Copyright year: 2014
Copyright holder: Ruhl et al.
License: This is an open access article distributed under the terms of the Creative Commons Attribution License, which permits unrestricted use, distribution, reproduction and adaptation in any medium and for any purpose provided that it is properly attributed. For attribution, the original author(s), title, publication source (PeerJ) and either DOI or URL of the article must be cited.
License URL: https://creativecommons.org/licenses/by/4.0/

Keywords: Bluegill, Lepomis, Reservoir, Zone, Southern Ohio

Funding: This work was funded by Ohio University. The funders had no role in study design, data collection and analysis, decision to publish, or preparation of the manuscript.

==============================
Reservoirs exhibit gradients in conditions and resources along the transition from lotic to lentic habitat that may be important to bluegill ecology. The lotic–lentic gradient can be partitioned into three functional zones: the riverine, transitional, and lacustrine zones. We measured catch frequency and length of bluegills (Lepomis macrochirus) captured along the periphery of these areas (i.e., in the littoral zone of each functional zone) for four small reservoirs in Southeastern Ohio during the summer months of three years. Catch frequency differed between zones for two reservoirs, but these differences were not observed in other years. There was no relationship between reservoir zone and either standard length or catch frequency when the data for all reservoirs were pooled, but we did observe a bimodal length distribution in all reservoirs. A combination of ecological factors including inter and intraspecific competition, predation intensity, management practices, limnology, and assemblage complexity may be mitigating bluegill distribution and abundance in reservoirs. Therefore, a functional zone (categorical) approach to understanding bluegill ecology in reservoirs may not be appropriate.

Introduction

Bluegill (Lepomis macrochirus) exhibit ontogenetic habitat shifts that coincide with shifts in foraging behavior in natural lakes. After hatching in the littoral zone, young-of-year bluegill migrate to the limnetic zone to feed on zooplankton (Werner, 1969). Once a larger body size has been attained, bluegill return to the littoral zone and feed opportunistically amongst macrophytes. After several years feeding in the littoral zone, larger bluegills shift back to a diet of zooplankton and move freely between the littoral and limnetic zones (Mittelbach, 1981). Shifts in bluegill diet and habitat use may be a result of a trade-off between maximizing foraging efficiency while minimizing predation risk (Werner & Hall, 1988). However, Wildhaber & Lamberson (2004) suggested an alternative hypothesis based on a hierarchical model of tradeoffs between prey availability and temperature in lakes. Regardless of the specific cause of the shift in bluegill habitat use (direct selection pressure via predation vs. indirect selection pressure from prey/habitat availability), it is an effective life history strategy (reviewed by Werner & Peacor, 2003).

The success of habitat switching as a life history strategy for bluegill may depend on a number of factors. For instance, basin morphometry may lead to differential recruitment success of bluegill amongst natural lakes where maximum depth, percent littoral area (Tomcko & Pierce, 2001), and lake surface area (Tomcko & Pierce, 2005) have all been linked to recruitment success. Littoral bluegill abundance is positively associated with habitat features such as the availability of woody debris (Newbrey et al., 2005), and native macrophytes (Theel & Dibble, 2008). Another important factor is the availability of zooplankton (Garvey & Stein, 1998); lakes with low productivity or high turbidity have low epilimnetic phytoplankton abundance which reduces zooplankton and thus decreases bluegill recruitment (Stein, DeVries & Dettmers, 1995). High abiotic turbidity in the photic zone is normally driven by physical processes such as wind mixing and flooding but also can be influenced by sympatric species that resuspend sediment (e.g., gizzard shad; Vanni et al., 2005) resulting in both direct and indirect density-dependent effects on bluegill recruitment via alteration in prey availability and/or capture success (Aday & Hoxmeier, 2003; Shoup et al., 2007). Indeed, protracted spawning by bluegill (Garvey, Herra & Leggett, 2002; Kaemingk et al., 2014) may be an adaptation to offset density-dependent effects caused by competition for prey (Partridge & DeVries, 1999; Michaletz, 2006; but see Leonard, DeVries & Wright, 2010).

Reservoirs exhibit gradients in the relative area of littoral vs. limnetic habitat (Thornton, 1990), zooplankton community composition (Bernot et al., 2004), and a suite of environmental variables including turbidity (Thornton, 1990) and available nutrients (Kennedy & Walker, 1990) vary along the lotic-lentic transition. Reservoirs can be divided into three functional zones based on these gradients (Fig. 1): the fluvial zone is the shallow unstratified portion that is heavily influenced by flooding and where well-mixed epilimnetic water is in direct contact with sediments, the transitional zone is weakly stratified and less influenced by flooding or sediment resuspension, and the lacustrine zone is the stably stratified lake-like area (adapted from Kimmel, Lind & Paulson, 1990).

Figure 1 Diagram of vertical and horizontal zonation in a stereotypical reservoir.

The curve ending in 4 °C represents a stereotypical summer thermocline in a deep reservoir.

Interpreting the ecological dynamics of reservoirs in the paradigm of functional zones along the lotic-lentic transition has been regularly applied to organisms that are at the whim of hydrologic conditions (reviewed by Ruhl, 2013), but to our knowledge has not been explicitly assessed in relation to more motile species such as fish. Additionally, the functional zonation paradigm for reservoirs typically describes open water rather than along the shoreline (littoral zone), despite the fact that differences in the mixing regime in open water may directly influence factors such as nutrient availability along the periphery. Because bluegill ecology is intimately linked to the conditions and resources in the limnetic as well as littoral zones, the functional zone paradigm may be particularly relevant and yield insight into broadscale differences in their ecology within reservoirs (i.e., both along the lotic-lentic gradient and between the littoral and limnetic zones). Specifically, we predicted that size and catch frequency may vary among functional zones because of differences that affect bluegill recruitment (i.e., their suitability for growth and reproduction, see above). We sampled bluegill abundance in the littoral zone throughout four different reservoirs to assess their use of the lotic-lentic transition, (in multiple years in some cases) during July and August when stable thermal stratification is normally strongest and therefore differences among functional zones may be at their peak.

Methods

Study sites

We sampled Dow Lake, Lake Hope, Lake Snowden and Fox Lake. These four reservoirs are located in close proximity to one another in the unglaciated hills of Southeastern Ohio and are managed by units of the Ohio Department of Natural Resources (Fig. 2, Table 1).

Figure 2 County map of Ohio highlighting the counties where trapping occurred (gray shading; Vinton and Athens) and location of the reservoirs (asterisks).

The bold line indicates the extent of glaciation.

Table 1 Basin morphometrics, fill date, and date of last bluegill stocking for each reservoir.

Variable	Dow	Fox	Hope	Snowden	
Catchment area (km2)	18.90	10.36	25.64	9.78	
Surface area (km2)	0.67	0.23	0.48	0.65	
Maximum depth (m)	9.5	6.0	6.5	10.0	
Mean depth (m)	1.62	1.28	1.31	2.45	
Volume (m3)	1,085,069	2,94,975	630,744	1,590,558	
Shoreline length (km)	11.27	3.86	9.18	11.91	
Shoreline development	5.49	3.21	5.29	5.89	
Maximum fetch (km)	2.00	0.71	1.07	2.87	
Fill date	1960	1968	1939	1970	
Last stocked with bluegill	1972	N/A	1979	1970	

Dow Lake (Stroud’s Run State Park) is used primarily for recreation, but also to mitigate flooding of the Hocking River downstream of Athens, Ohio. This reservoir was filled during 1960 and the watershed is composed of minimally disturbed hills, woodland, and open fields. Throughout the reservoir, the littoral zone has been modified via the felling of shoreline trees and addition of brush piles to coves during the early 2000s (M Greenlee, District 4 ODNR Biologist, pers. comm., 2010).

Lake Snowden was filled during 1970. The reservoir previously supplied drinking water to the surrounding community, but currently is used for flood control, hatchery water supply, and recreational activities. The watershed consists of rolling hills, agricultural fields, and woodlots while the shoreline habitat includes submerged trees, overhanging brush and abundant submerged macrophytes.

Fox Lake was filled during 1968 and the watershed is composed of rolling hills, agricultural fields and woodlots. High sedimentation rates in the riverine zone have resulted in poor angler access to the reservoir and, consequently, submerged macrophytes were removed during the mid-1990’s in order to increase flow and accessibility to the riverine zone (M Greenlee, District 4 ODNR Biologist, pers. comm., 2010). These efforts have not been successful in improving angler access and dredging to manually remove sediment is impractical for this reservoir.

Lake Hope is located within the Zaleski State Forest and was filled during 1937. The watershed is composed of mature second growth forest scattered with abandoned pit and shaft coal mines. The reservoir has abundant invasive emergent macrophytes (primarily fragrant water lily, Nymphaea odorata) mixed with a variety of other emergent and submerged macrophytes around the periphery.

Bluegill are not regularly stocked into any of the reservoirs (Table 1). Rainbow trout (Oncorhynchus mykiss, mean length 303 mm, 2001–2011) are stocked into Dow Lake every April. All four reservoirs are stocked yearly or in alternating years with channel catfish (Ictalurus punctatus, mean 221 mm) during the Fall. Lakes Snowden and Hope are stocked with saugeye (Sander canadensis × Sander vitreus, mean 31.5 mm) every year during the Spring. Fish are normally stocked into the reservoirs in close proximity to the boat launch, meaning that stocked fish are introduced into the riverine zone at Dow Lake and Fox Lake, the transitional zone at Lake Snowden, and the lacustrine zone at Lake Hope.

Sampling regime

The reservoirs were sampled using shore-line traps during three different years, but only Dow Lake and Lake Hope were repeatedly sampled (Table 2). Trapping was conducted during July and August in all years, but the number of weeks during which trapping occurred varied by year. All trapping was conducted using a randomized block design both within and among reservoirs, thereby minimizing the likelihood of a temporal effect among reservoirs or reservoir zones within a given year. Sampling methods were in accordance with Ohio University IACUC protocols and Ohio Department of Natural Resources Permit #464.

Table 2 Summary of the number of sampling sites and the total catch for each reservoir, zone, and year that sampling occurred.

Reservoir/year	Zone	#Sites	Total catch	
Dow 2006	Riverine	4	57	
Transitional	4	22	
Lacustrine	8	67	
Fox 2006	Riverine	5	72	
Transitional	2	19	
Lacustrine	2	27	
Dow 2007	Riverine	17	104	
Transitional	11	103	
Lacustrine	12	92	
Hope 2007	Riverine	4	31	
Transitional	5	47	
Lacustrine	6	60	
Snowden 2007	Riverine	3	25	
Transitional	6	59	
Lacustrine	7	40	
Hope 2008	Riverine	4	19	
Transitional	6	47	
Lacustrine	10	43	
Notes.

Raw data were used for statistical analysis (not the means presented here).

At each trapping site, pairs of oval traps (Promar ‘large’ 81 × 50 × 30 cm (1 cm mesh size and 12 cm minimum tunnel diameter) and ‘extra-large’ 91 × 62 × 50 cm (2.5 cm mesh and 15 cm tunnel diameter)) were positioned approximately 2 m from one another with trap entrances positioned parallel with the shoreline. The distance that the traps were positioned from shore was dictated by the slope of the shoreline; in order to avoid drowning concurrently caught turtles. The traps were positioned such that a small portion of trap protruded from the surface. Each site used two ‘large’ traps during the 2006 sampling season. During the 2007 and 2008 sampling season each site had one ‘large’ and one ‘extra-large’ trap. ‘Extra-large’ traps were introduced during 2007 and 2008 to ensure that we were not excluding larger bluegills and thus validate the 2006 size data. Trapping sites were located at approximately equal intervals around the periphery (littoral zone) of each reservoir. Upon arriving at the pre-determined trapping location, the exact positioning of the traps was again dictated by the slope of the shoreline. Each trap was baited with commercially available dip bait (Premo brand ‘original super-sticky dip bait’) hung inside the trap in a cheesecloth bag and positioned in as flat a position as the shoreline allowed in order to keep turtles and fish from getting under the trap or causing the trap to shift into deeper water. We checked traps every 24 h for five days and recorded species and standard length of all fish and then released them immediately.

Analysis

We determined the transitional zone area and the size of the riverine and lacustrine zones a posteriori for each reservoir and each year. We defined the transitional zone as the area of the reservoir where the presence of thermal stratification fluctuated due to weather conditions (wind and flooding) during the sampling period. Therefore, the transitional zone begins at the point when a well-mixed epilimnion and a metalimnion are present outside of the thalweg (if present) and continues until underflows terminate into interflows through the metalimnion (Fig. 1).

Length and catch frequency (the total number of fish caught during a five-day period for each site) could not be normalized for all groups, so comparisons among reservoir zones (i.e., within each reservoir) were conducted using Kruskal–Wallis tests and a priori Mann–Whitney U tests. The same tests were used when comparing catch frequency among reservoir zones for all reservoirs combined, but one-way ANOVA with post-hoc Tukey tests were employed to compare the standard length among zones for all reservoirs combined. Although the length data were not normal, ANOVA is robust for non-parametric data at sample sizes of 30 or greater for each group when the model is balanced. For our unbalanced model, normality can generally be assumed at sample sizes greater than 100 per group and our sample sizes are approximately 300 per group. We compared the catch frequency of small vs. large bluegills among reservoir zones for the pooled data (all reservoirs combined) using a two sample Kolmogorov–Smirnov test. All statistics were performed using SPSS 12.0.

Results

Standard length

Standard length only varied by reservoir zone for Dow Lake during 2006. In that case, bluegill caught in the transitional zone were smaller than those caught in the other zones (riverine: Mann–Whitney, U = 412, p = 0.019; lacustrine: U = 431.5, p = 0.004), but this result was not seen during 2007 (Fig. 3). When the length data from all reservoirs was combined, there were no differences among zones (one-way ANOVA, F(2, 822) = 0.053, p = 0.921).

Figure 3 Box-plot of bluegill standard length between zones at Dow Lake during 2006 and 2007.

Bluegills caught in the transitional zone during 2006 were significantly smaller than those caught in the riverine (Mann–Whitney, U = 412, p = 0.019) or lacustrine (U = 431.500, p = 0.004) zones, but this result was not observed during 2007. Box represents first and third quartiles, whiskers positioned at ±2 SD, horizontal line is the mean.

Catch frequency

There was no difference in the catch frequency of bluegill among reservoir zones for any of the reservoirs (Table 3). Catch frequency did not vary among reservoir zones when the data from all reservoirs was pooled either (Kruskal–Wallis, χ2 = 1.094, p = 0.579). Because the distribution of lengths was bimodal for all reservoirs in all years, we split the dataset at the saddle of the distribution (> / <8.5 cm, Fig. 4) and asked if the number of small or large bluegills varied over reservoir zone for each reservoir. Only in Lake Hope were small bluegills encountered more often in the transitional zone than in the fluvial zone during 2008 (Mann–Whitney, U = 357, p = 0.019, Fig. 5), but this result was not observed during the previous year. Combined for all reservoirs, there was no difference in the catch frequency of both small and large bluegills among zones (Kruskal–Wallis; small: χ2 = 2.285, p = 0.319; large: χ2 = 0.406, p = 0.816). Additionally, there was no relationship between the catch frequency of small vs. large bluegills among reservoir zones (Kolmogorov–Smirnov, Z = 1.083, p = 0.192).

Figure 4 Histogram of bluegill standard lengths for all reservoirs combined.

The dashed line indicates the saddle in the distribution at 8.5 cm where the data was bifurcated into “small” and “large”.

Figure 5 Catch frequency of small bluegills between zones in 2007 and 2008 at Lake Hope.

Box-plot of catch frequency of small bluegills (<8.5 cm) among zones during 2007 and 2008 at Lake Hope. Catch frequency in the riverine zone was significantly lower than in the transitional zone in 2008 (Mann–Whitney, U = 357, p = 0.019), but there were no significant difference among zones in 2007. Box represents first and third quartiles, whiskers positioned at ±2 SD, horizontal line is the mean.

Table 3 Results of a Kruskal–Wallis analysis of catch frequency between reservoir zones for each reservoir and year.

Lake	Year	χ 2	p-value	
Fox	2006	2.881	0.237	
Dow	2006	5.094	0.078	
Dow	2007	0.550	0.760	
Snowden	2007	0.793	0.673	
Hope	2007	0.812	0.666	
Hope	2008	1.832	0.400	

Discussion

Bluegill populations are influenced by a variety of factors including both abiotic factors such as turbidity (Stein, DeVries & Dettmers, 1995) or temperature (Wildhaber & Lamberson, 2004) and biotic factors such as prey availability (Garvey & Stein, 1998; Hoxmeier, Aday & Wahl, 2009) or predators (Werner & Hall, 1988); all of these factors vary dramatically among reservoir zones as a simple function of stratification regime (as well as other factors such as nutrient loading, water retention time, etc.) (Kimmel, Lind & Paulson, 1990). However, few differences in bluegill among reservoir zones were observed in our study. Size of bluegill differed among zones at Dow Lake during 2006, but this result was not observed during 2007. Similarly, small bluegills were caught more frequently in the transitional zone at Lake Hope during 2008, but not during 2007. When the data from all reservoirs was pooled, there were no differences in either size or catch frequency among reservoir zones, suggesting that habitat partitioning may be based on different criteria in reservoirs (Gelwick & Matthews, 1990; Eggleton et al., 2005) than has previously been described for natural lakes (e.g., Werner et al., 1977).

The lack of repeatability in our findings among years may be indicative of the true nature of reservoirs as a habitat for bluegill. Resources and conditions within a reservoir may be dependent on prevailing weather patterns (Lienesch & Matthews, 2000; but see Edwards et al., 2007), inputs from the watershed (Gido et al., 2002; Vanni et al., 2005) and presence of certain species (e.g., gizzard shad; Vanni et al., 2005). All of these variables can fluctuate dramatically year to year and cause shifts in prey availability (Betsill & van den Avyle, 1994) and predation intensity (Jackson & Noble, 2000). Additionally, the artificial, managed nature of reservoirs creates dynamics environments where water levels and habitat availability/suitability (Collingsworth & Kohler, 2010) and stocking of competitors (Leonard, DeVries & Wright, 2010) and/or predators may vary yearly. Therefore, while size and catch frequency of bluegill may differ by reservoir zone at times (as we observed at Dow during 2006 and Hope during 2008), they are likely influenced by other factors as well, which may have disrupted our ability to consistently detect differences among zones.

Bluegill spawning behavior also may influence the detectability of differences in length and catch frequency among reservoir zones (if they exist). Bluegill spawning is condition-dependent for males (males in better physical condition spawn first; Cargnelli & Neff, 2006), which results in protracted spawning (spawning over an extended period; Kaemingk et al., 2014). Given the differences in prey availability among reservoir zones (Betsill & van den Avyle, 1994), protracted spawning may be more prevalent in reservoirs than in lakes and could cause behavioral plasticity in habitat use that is difficult to detect using standard techniques (e.g., trapping, netting, or electro-shocking). That is, if bluegill spawning occurs over a wider range of times in reservoirs, population wide shifts in habitat use would be similarly spread over a longer duration and differences among zones, which may be important to bluegill, may also be difficult to detect. This is supported by Jolley, Edwards & Willis (2009) who found that the timing of spawning in bluegill varied among nearby reservoirs and among years in the same reservoirs.

The size structure of the bluegills we caught by trapping (all reservoirs combined) was bimodal and somewhat positively skewed. The positive skew was determined by the uniformly smaller bluegill that freely travel through the traps without being caught while the right tail extends because larger individuals are rare. The saddle in the size distribution at approximately 8.5 cm is more intriguing. Bluegills < 10 cm (except planktivorous larvae) are normally found in the littoral zone of lakes because this area provides the greatest protection from predation (Werner & Hall, 1979). It may be that in our study, bluegill move away from the shoreline reservoir-wide at a much smaller size in reservoirs then in natural lakes, but we believe this is unlikely given the differences in ‘offshore’ conditions and resources among reservoir zones. Likewise, it is possible that the saddle of the distribution represents two different age classes, but this is also unlikely given the variation in growth rates observed in bluegills among reservoirs (Jackson, Quist & Larscheid, 2008) and their protracted spawning behavior. More likely, the saddle is a result of size selective predation by largemouth bass (Olson, 1996) or other piscivores such as saugeye. Because these piscivores are gape limited, bluegills over approximately 10 cm (Werner & Hall, 1979) are at lower risk of predation than smaller bluegills (Santucci & Wahl, 2003). Therefore, the saddle may represent the point at which size-specific mortality of bluegill caused by predation (Mittelbach & Persson, 1998) starts to decline in Southeastern Ohio reservoirs.

Lastly, another factor that may have contributed to our results is that our methodology did not detect temporal variation within a reservoir. Because trapping occurred during the course of a few weeks for each reservoir, differences in catch frequency or size among zones as a result of behavioral plasticity during ontogeny may be diluted. However, Gelwick & Matthews (1990) suggest that there is little temporal variation in littoral fish assemblages of reservoirs relative to lakes because these assemblages are ‘evolutionarily short-lived’. That is, because a given reservoir has not existed long in evolutionary time, fish assemblages may not exhibit the same patterns seen in natural lakes which have existed for many years. Our results seem to support this conclusion given that we only saw differences in the oldest of the reservoirs we sampled. Similarly, anthropogenic factors such as intensive stocking (Gelwick & Matthews, 1990) or the maintenance of a community dominated by a small number of species (Eggleton et al., 2005) may contribute to a decrease in temporal variation in habitat use in reservoirs.

In this study, bluegill generally did not differ in size or catch frequency among reservoir zones in four Southeastern Ohio reservoirs. This result, although unexpected due to the broad differences in habitat characteristics among reservoir zones, may be caused by a combination of factors including prey availability relative to predation intensity in reservoirs, management practices, limnology, and assemblage complexity. Kaemingk et al. (2014), working with limnetic bluegills, hypothesized that similar factors may regulate the timing and duration of spawning behavior, which should have reinforced differences in bluegill abundance among reservoir zones. Despite the fact that significant differences were found using the same approach with aquatic turtles (Ruhl, 2013), we did not detect differences in bluegill size or catch frequency among zones. Thus, it is likely that a categorical (functional zone) approach to detecting differences in bluegill ecology within a reservoir is not appropriate.

We wish to thank all of the undergraduate assistants who participated in sampling, the members of the Roosenburg lab for their helpful suggestions, and Drs. Warren J.S. Currie and K Cuddington for their assistance. Two anonymous reviewers provided comments that improved an earlier version of this manuscript.

Additional Information and Declarations

Competing Interests

Author Contributions

Animal Ethics

Field Study Permissions

The authors declare there are no competing interests.

Nathan Ruhl conceived and designed the experiments, performed the experiments, analyzed the data, contributed reagents/materials/analysis tools, wrote the paper, prepared figures and/or tables, reviewed drafts of the paper.

Holly DeAngelis and Abigale M. Crosby analyzed the data, wrote the paper, prepared figures and/or tables, reviewed drafts of the paper.

Willem M. Roosenburg analyzed the data, wrote the paper, reviewed drafts of the paper.

The following information was supplied relating to ethical approvals (i.e., approving body and any reference numbers):

This study was conducted in accordance with Ohio University’s IACUC guidelines.

The following information was supplied relating to field study approvals (i.e., approving body and any reference numbers):

Ohio Department of Natural Resources #464.

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
