# Peer review of "Applying a reservoir functional-zone paradigm to littoral bluegills: differences in length and catch frequency?"

_PeerJ, doi:10.7717/peerj.528_

## Round 0.1 · original submission · Major Revisions

Your manuscript has been reviewed by two independent referees. I request that you take into account all their comments.

·

Basic reporting

“Bluegill” is a species. “Bluegills” are individuals of a species. You use the terms interchangeably, particularly in your introduction; be certain of which term you intend to use.

“In” refers to location (“in” the river); “during” refers to time (“during” the 1970s, NOT “in” the 1970s)

Please review the appropriate usage of commas and hyphens; there are numerous places in the manuscript where they are used incorrectly, or not used when they should be. This makes the manuscript cumbersome to read.

Particularly in the methods section, you alternate between active and passive voice: “We sampled the river mouth” versus “flow meters were mounted.” Choose one style and be consistent throughout. Unless the journal has a convention you need to adhere to, active voice is generally more interesting to read.


Please search for additional references about age-0 bluegill by M. A. Kaemingk; I believe you may find them useful for this and other citations.

Experimental design

No Comments.

Validity of the findings

No Comments.

Additional comments

I enjoyed reading this manuscript. I commend the authors on attempting to publish what many would consider “non-significant” results. Overall, the manuscript is well written from a structural standpoint. However, I believe the manuscript, in its current form, is in need of some improvements and a good deal of polish to be acceptable for publication.

General comments:

“Bluegill” is a species. “Bluegills” are individuals of a species. You use the terms interchangeably, particularly in your introduction; be certain of which term you intend to use.

“In” refers to location (“in” the river); “during” refers to time (“during” the 1970s, NOT “in” the 1970s)

Please review the appropriate usage of commas and hyphens; there are numerous places in the manuscript where they are used incorrectly, or not used when they should be. This makes the manuscript cumbersome to read.

Particularly in the methods section, you alternate between active and passive voice: “We sampled the river mouth” versus “flow meters were mounted.” Choose one style and be consistent throughout. Unless the journal has a convention you need to adhere to, active voice is generally more interesting to read.

Your reference style is inconsistent. Lines 283 and 322 use periods after initials. There are also many places where a single space or a double is used after a period; be consistent.

The last paragraph of the discussion is unimpressive. I believe you should move the “recommendations for future research” to their own paragraph (second-to-last paragraph), and finish with a more compelling paragraph. If you want a management “hook”, go that direction. If a reservoir ecology hook, go that direction.

Specific Comments:
1st line of Abstract: reservoir’s don’t “possess” anything. They do “exhibit” things.

7th line of Abstract: change “repeatable” to “observed”.

Last sentence of Abstract: is 4 lines long and confusing. Change to “… in reservoirs. This may necessitate…”.

Line 7: “young-of-year” what? Change to “young-of-year bluegill”.

Line 8: change “obtained” (which implies possession) to “attained”.

Line 8: change “the fish” to “bluegills”.

Lines 11, 15: Inconsistent phrasing. Change all uses to “bluegill diet/habitat use” rather than “diet/hab use by bluegill”.

Line 16: change “or” to “versus”. Insert “selection” between “indirect” and “pressure”.

Line 20: change “between” to “among”. Use between when comparing 2 things, among when comparing 3 or more. There are numerous incorrect uses of “between”; I have tried to locate them all, but may have missed one or more.

Lines 22-24: Bluegills are positively associated with woody debris and macrophytes, not the other way around.

Lines 31-33: Please search for additional references about age-0 bluegill by M. A. Kaemingk; I believe you may find them useful for this and other citations.

Line 33: “Partridge”.

Line 46: you use “functional-zonation scheme” here, and “functional-zone paradigm” on Lines 51 and 229. Be consistent.

Line 48: remove hyphen in “open-water”.

Line 49: Because it’s poor grammar, never begin a sentence with “Because”. In this sentence, delete “because” and add “thus” or “therefore” after the comma.

Line 62: You did not trap lakes. You trapped bluegills.

Line 62: improper use of a semicolon; the phrase after the semicolon must be a complete sentence.

Line 66: “during” 1960.

Lines 66, 70, 75, 81: inconsistent descriptions of lake construction and filling. Unless God works for the OH DNR, Lake Snowden was not created. Don’t use “Lastly” when you haven’t use Firstly, Secondly, or Thirdly. Structure each of these paragraphs the same.

Line 84: common name for Nymphaea odorata?

Line 95: change “over” to “during”. Over implies space or position, whereas “during” implies time.

Line 96: did you perform “sampling” or “trapping”? Be consistent.

Line 121: change “over” to “during”.

Line 122: “five-day period”.

Line 127: “data were”. “Data” is plural.

Line 131: “data are”.

Lines 134-152: too many digits used in many of your test statistics and p-values. Review the use of significant figures. Also, three instances of “data was”; change to “data (or dataset) were”.

Line 145: change to “varied among reservoir zones”.

Line 149: change “between” to “among”.

Line 150: missing a leading “0” for your large Chi-square statistic.

Line 159: change “between” to “among”.

Line 161: change “repeatable” to “observed”.

Line 162: “data were”.

Line 173: I’m sure Dr. Mike VanDenAvyle would appreciate it if you used proper capitalization. Also on Line 184 and in references on Line 245.

Line 184: change “between” to “among”.

Line 185: remove hyphen from “habitat-use”.

Line 188: change “time-frame” to “period” or “duration”.

Line 192: change “through” to “by”.

Line 193: change “While” (which implies time) to “Although”. Which mesh size? “smaller fish and larger individuals” is poorly written.

Line 195: change “about” to “approximately”.

Line 198: change “feel” to “believe”.

Line 199: change “between” to “among”.

Line 209: see comment for Line 49.

Line 209: change “over” to “during”.

Line 219: remove hyphen from “habitat-use”.

Line 220: change “usually” to “generally”.

Line 220: change “between” to “among”.

Line 221: change “while” to “although”.

Line 222: change “between” to “among”.

Line 225: change “tease apart” to “elucidate” or “evaluate” or “determine”.

Lines 227, 228: none of what you describe is “easily manipulated”. Change this phrasing.

Line 229: change “while” to “although”.

Line 229: change “work for” to “apply to”.

Lines 286, 303: “DeVries”.

Table 1: All of your tables are poorly formatted. Inconsistent decimal places, decimals don’t align within columns, km2 and m3 not properly typeset, numbers wrapping between rows. Moreover, I would transpose the whole table so that the reservoirs are in columns and the variables in rows. Comparisons among columns are much easier for most readers to interpret.

Table 2: Table caption is poorly written; revise it. Column 1 is reservoir/year, not site/year, and should be two columns anyway. If you’re going to provide means, also provide some measure of variance (variance, SD, SE, 95% CI’s, etc.). Too many decimal places in means; only 1 decimal place is necessary. “raw data were”.

Table 3: “Snowden” wraps on two rows. Excessive decimal places in Chi-square stats and p-values.

Figure 1: Use an actual degree symbol, not a superscript zero.

Figure 2: Resolution of this figure is poor. Re-make using GIS if possible.

Figure 3: standard length of what? Change “in” to “during”. See previous comments about decimal places and significant figures for stats and p-values. Resolution of this figure is also poor.

Figure 4: Standard length? Total length? Eye-fork length? Y-axis should be on left side of figure.

Figure 5: define “small” in your caption. Change “between” to “among”. Change “in” to “during”. “there was no significant difference”.

Reviewer 2 ·

Basic reporting

Overall, this manuscript is a straightforward presentation of bluegill trapping data from 4 Ohio reservoirs. The authors found few differences in bluegill size or catch rates among the 3 reservoir functional zones. The Introduction is well written with up-to-date scholarship, with my only suggestion being to more explicitly outline the objectives with specific hypotheses. For example, instead of stating that they would see differences among zones, state what they hypothesized those differences might be (smaller fish in riverine vs. lacustrine, or something).

Additional information and corrections are needed in the Tables and Figures.
Table 1. How does Hope have a volume of 4m3? Typo?
Table 3. “n” in Snowden dropped a line. Fix table alignment.
Figure 3. Indicate which zones are which as in Figure 5. Is zone 1 the riverine zone, 2 the transitional, and 3 the lacustrine? Indicate either on the figure or in the figure caption. Figure caption uses “fluvial” and other parts of manuscript use “riverine”. Are these meant to refer to the same thing? If so, pick one and be consisten throughout. Also state that it is a boxplot and indicate what the whickers represent (75% quartile? Standard error of the mean? or something else?). The graph really doesn’t depict the difference among zones in 2006. Perhaps rearrange the graph so you can better show this difference. Maybe split the years apart (all 3 2006 zones one the left and all 3 2007 zones on the right). Then it would be easier to see the smaller fish in the transitional zone.
Figure 5. in caption, replace “between zones” with “among zones”. Include information about the symbols in the plot. Means? 50% quantiles? Standard deviations?

Experimental design

The science and statistics are sound. The trapping methods need more information including how deep the traps were set, how far from the shoreline were they set, and the types of habitat that they were set in. For example, were they all set at the same depth and in macrophyte beds?

Validity of the findings

The findings are straightforward and appropriately stated.

Additional comments

Line 6. Include the scientific name of bluegill. It’s in the abstract, but it needs to be stated in the Introduction too.
Line34. Rephrase away from “…, there are gradients….”
Lines 43-59. I suggest separating out the objectives of this study. As it stands, this paragraph contains both the rational for the conceptual framework of the study and also the objectives. Also, the objectives could be more explicitly stated and explained. I would include more specific hypothesis beyond the very general statement catch frequency may vary. This paragraph containing the objective would be strengthened by stating how they may vary (i.e., include specific predictions).
Lines 62-69. This paragraph contains an overview of the four reservoirs and a description of on eof the reservoirs. The other reservoirs are described in their own paragraphs. I suggest either describing all of the reservoirs in one paragraph or devote an entire paragraph to each and every reservoir.
Paragraph beginning line 102. How deep were the traps places? How far from the shoreline?
Line 127. “Although the length data was not normal…” a citation is needed for this statement. I don’t think it is common knowledge that we can use anova for non-normal data if the sample size is greater than 100. Also, change “data was” to “data were”.
Line 177-179. Seems like the “zones” themselves change seasonally and from year to year, especially the “transitional zone”
Line 222. “…differences in habitat characteristics between reservoir zones…” I understand that the broader “zones” are different, but I can’t find enough information to determine if the places that were sampled in this study had different habitat characteristics. The depth of traps was not provided, but I assume that they were set a similar depth. So perhaps the habitat at that particular depth is pretty much the same even in different “zones”. Or maybe I’m missing something here. Were the littoral areas of each reservoir zone sampled? The manuscript could be improved by better explaining the sampling specifics.

---

## Round 0.2 · Minor Revisions

Please check the points raised by referee 1.

·

Basic reporting

No Comments

Experimental design

No Comments

Validity of the findings

No Comments

Additional comments

Comments to Authors:

As before, I enjoyed reading this manuscript. I believe the manuscript is improved from the previous version; the authors did a fine job of incorporating suggestions from both reviewers. I have only a few minor editorial suggestions that persisted from the first version, plus a couple of specific suggestions.

General comments:
“In” refers to location (“in” the river); “during” refers to time (“during” the 1970s, NOT “in” the 1970s).

Change “between” to “among”. Use “between” when comparing 2 things, “among” when comparing 3 or more (“among reservoir zones”, like you used on Line 149, 190). There are still numerous incorrect uses of “between”.

Specific comments:
Lines 106-121: Though I did not comment on it during the first round, I appreciate the increased detail describing your field methods.

Line 219: I believe you have size-specific mortality, not necessarily age-specific mortality.

Line 238: change “be at play in determining” to “regulate”

Lines 240-243: I recommend against ending your bluegill paper with a sentence about turtles (even if it was your own work). I suggest moving the sentence elsewhere, or at the very least, restructuring it to something like the following: “Despite the fact that significant differences were found using the same approach with aquatic turtles (Ruhl, 2013b), we did not detect differences in bluegill size or catch frequency among zones. Thus, it is likely that a categorical (functional zone) approach to detecting differences in bluegill ecology within a reservoir is not appropriate.”

Line 333: Thornton reference isn’t formatted appropriately.

Figure 3: caption indicates gray and white bars, but figure doesn’t depict them as such. You describe the box and whiskers, but not the line in the box – I assume it’s the mean, but some graphical programs display the median.

Figure 5: same as Figure 3. The outlier symbols are very difficult to see.

Table 2: your note below the table is no longer relevant and can be removed, though I appreciate you changing “data was” to “data were” :).

---

## Round 0.3 · accepted · Accept

The authors have addressed all the points